# Anatomical, Histological, and Morphometrical Investigations of the Auditory Ossicles in *Chlorocebus aethiops sabaeus* from Saint Kitts Island

**DOI:** 10.3390/biology12040631

**Published:** 2023-04-21

**Authors:** Cristian Olimpiu Martonos, Alexandru Ion Gudea, Ioana A. Ratiu, Florin Gheorghe Stan, Pompei Bolfă, William Brady Little, Cristian Constantin Dezdrobitu

**Affiliations:** 1Faculty of Veterinary Medicine, University of Agricultural Sciences and Veterinary Medicine Cluj-Napoca, 400372 Cluj-Napoca, Romania; 2School of Veterinary Medicine, Ross University, Basseterre P.O. Box 334, Saint Kitts and Nevis; 3Faculty of Medicine and Pharmacy, University of Oradea, 410087 Oradea, Romania

**Keywords:** auditory ossicles, green monkey, morphology, morphometry, histology

## Abstract

**Simple Summary:**

The present paper explores a series of macromorphological, micromorphological, and morphometrical aspects of the auditory ossicles of the green monkey. The particular aspects of the gross morphologies of the three ossicles are described in detail, comparing data with those on other primates and non-primates, underlining the most important features. This study adds some histological insights (never before described to the present date) on all three adult auditory ossicles and supplies a series of metrical data (raw and other complex interpretations of data) to complete the knowledge of the functional anatomy of the lever system in ear ossicles.

**Abstract:**

Otological studies rely on a lot of data drawn from animal studies. A lot of pathological or evolutionary questions may find answers in studies on primates, providing insights into the morphological, pathological, and physiological aspects of systematic biological studies. Our study on auditory ossicles moves from a pure morphological (macroscopic and microscopic) investigation of auditory ossicles to the morphometrical evaluation of several individuals as well as to some interpretative data regarding some functional aspects drawn from these investigations. Particularities from this perspective blend with metric data and point toward comparative elements that might also serve as an important reference in further morphologic and comparative studies.

## 1. Introduction

The existence of the green monkey (order: Primate, suborder: Haplorhini, infraorder: simiiformes, family: Cercopithecidae, and tribe: Cercopithecini) (*Chlorocebus aethiops sabaeus*) in the federation of Saint Kitts and Nevis is linked to the arrival of the first French colonists more than 300 years ago. They are a non-endangered, Old World non-human primate species.

Normally, this species is specific to Sub-Saharan Africa, but it reached several islands of the West Indies with the first slave ships. With no natural predators and with protection from specific African pathology, the population of monkeys on Saint Kitts thrived, reaching an estimated more than 45,000 individuals by the present date, becoming a significant problem for farmers in St. Kitts and complicating most agricultural activities [1,2,3].

Due to the size of the island and the high populational pressure of the monkeys, authors believe that a controlled yearly reduction of 5000 individuals would be necessary to limit this damage to agriculture and the resultant loss in gross domestic product [4].

The invasive nature of the African green monkey, its negative environmental impact associated with a growing population abundance, and the animal’s unique morphology have made it a prime target for use in research focused on non-human primates [2] and other phylogenetical or biomedical investigations [1,5].

Phylogenetically related to humans, this species shares many important features of our pathophysiological processes [6]. Although the list of successful research studies utilizing the green monkey is extensive, a few important examples include those that focus on pathogens such as *Yersinia pestis* [7], *Klebsiella pneumoniae* [8], *Staphylococcus aureus* [9], *Mycobacterium tuberculosis* [10], and several viruses [11], many immunological studies [12], and experimental models for cardiac transplants or cardiovascular pathology [1].

A series of studies concerning auditory ossicles (the incus, malleus, and stapes) in primates are found in the literature [5,13,14,15,16,17,18,19,20,21,22,23,24], especially from a comparative perspective with humans. Based on morphological features, the ossicular assembly is categorized into the Old World primate type, the New World Primate, and the Prosimian type [15,16,25].

The three auditory ossicles are located in the cavity of the middle ear (*Auris media*) of primates and play an important role in the transmission and modulation of soundwaves from the tympanic membrane to the internal ear components (*Auris interna*) [5,22,26,27].

Even though, in many ways, there are strong phenotypic similarities between *Chlorocebus aethiops sabaeus* (the green monkey) and *Homo sapiens* (humans), to the best of our knowledge, the scientific literature lacks significant data to document the anatomy of the auditory ossicles of the aforementioned species.

The present study aims to characterize, from an anatomical, histological, and morphometrical perspective, the auditory ossicles of the green monkey (*Chlorocebus aethiops sabaeus*).

## 2. Materials and Methods

### 2.1. Material of Animal Origin

The biological material originated from six fully mature Green African Monkey (*Chlorocebus aethiops sabaeus*) which were part of an IACUC approved terminal surgical study conducted by the BSF—Behavioral Science Foundation Saint Kitts and Nevis and were kindly donated to Ross University School of Veterinary Medicine after the endpoint Carcasses were weighing between 3.3 to 5.2 kg, (eviscerated weight).

### 2.2. Anatomical Image Collection and Processing Technique

After the identification of the external auditory canal, a stepwise removal (with a bone rongeur and manually) of the bony part was performed until the tympanic membrane was identified. Once the tympanic membrane was identified, the manubrium of the malleus became visible at the level of the umbo. With the help of a scalpel blade, the tympanic membrane was detached, and the middle ear compartment was entered. After the approach into the tympanic cavity, the presence of the ossicular chain was revealed: the malleus, the incus, and the stapes (in the lateral and medial directions).

For easier exposure of this cavity, as well as for better visualization of the auditory ossicles, the area was irrigated with normal saline and hydrogen peroxide. After removing the debris and fine bone splinters, the auditory ossicles were extracted with gentle dissection, and the harvested ossicles were preserved in 10% formaldehyde. Images were acquired with an Olympus SZX7 magnifying glass with an attached DP27 Camera, with cellSens Standard Imaging Software and were later processed with Adobe Photoshop for fine contrast and background processing. Further digital measurements and evaluations of the bony pieces were made using a scaled surface and ImageJ software at the Anatomy Department of the Faculty of Veterinary Medicine, Cluj-Napoca, Romania.

### 2.3. Histological Technique

After dissection, the malleus, incus, and stapes bones were fixed in 10% neutral buffered formalin for 2 days, stored in 70% alcohol, and then isolated, before being placed in a labeled cassette between biopsy pads. The tissues were then immersed in DeltaFORM decalcifier (Delta Medical Inc., Brookfield, WI, USA, lot 0403.423) for two hours. The histology cassettes were removed from the decalcifier and rinsed in tap water and then loaded into a Tissue-Tek E300 processor and processed during a standard processing cycle (60 min steps). The processed cassettes were embedded and cut at 5 µm with a Leica RM 215 microtome using Tissue-Tek Feather A35 low-profile blades. Some surface decalcification was needed for some sections as the initial decalcification (a 1 h, slow process) was incomplete. The slides were deparaffinized and stained with Hematoxylin and Eosin (HE), dehydrated, and cover-slipped with mounting media. The slides were examined using an Olympus light microscope (BX 43). Images were captured with a DP 26 digital camera and the cellSens Standard image analysis software for measurements.

### 2.4. Morphometrical Approach

Measurements of the ossicles were attained following a standardized and widely-accepted systematic approach, which has been well described by previous researchers [5,28] in otology studies of humans and other primates. Previous studies indicate that measurements of the malleus are the most relevant from a metrical perspective; however, those of the incus and stapes are less reliable [22], especially when targeting sex or side differentiation.

Special emphasis is placed on some measurements that are considered to be functional lengths (Table 1, Table 2 and Table 3) [5] (in the case of the malleus and incus mainly) that may be used in the calculations of the lever ratios in the middle ear to elucidate and describe the mechanical advantage associated with the lever system in the ear.

For some area calculations, the elliptic tool from the ImageJ application was used to overlap the tool onto a digital image and thus automatically calculate the surface of the measured surface.

The standardized measurements and the measurement protocols have been previously described for each of the ossicles [5]. All measurements were taken by one author using the same method of measurement and calibration in ImageJ and the same calibration scale over a short period of time—attempting, in fact, a type A uncertainty data repeatability test—and all photographs were taken by another author (to reduce biases). Measurements were taken from seriate images and from different angulations. This approach made some measurements not possible due to the different presentation of an anatomical piece. Data were recorded by hand and then imported into a Google calculation sheet (allowing the direct calculation of indices). Data were recorded separately using the identification of an individual, sex (male/female), and side (right/left). Statistical processing was carried out with Google Sheets’ simple statistics feature, the XL Miner Analysis Toolpack add-on 2023 (FrontlineSolvers—www.solver.com (accessed on 2 February 2023), the XLSTAT Clud app, and the online website of Social Science Statistics (https://www.socscistatistics.com/ (accessed on 2 February 2023)).

For the incus, the next series of measurements were taken [5].

For the stapes, we followed the same protocol [5].

## 3. Results

### 3.1. Malleus

The malleus (*malleus*) (Figure 1) is the most lateral ossicle in contact with the tympanic membrane. The ossicle comprises three known parts: the head of the malleus (*Caput mallei*), the neck of the malleus (*Collum mallei*), and the handle of the malleus (*Manubrium mallei*).

The head (1) is quite evident, round, tuberous, and slightly elongated in the mediolateral direction. The lateral surface, which is slightly convex, shows the existence of numerous surface accidents that mark the insertion of the superior suspensory ligament (the superior malleal fold) and the lateral suspensory ligament (the lateral malleal fold) that attach the malleus to the walls of the tympanic cavity.

The medial surface, which is slightly concave, shows the articular surface for its articulation with the incus (*Facies articularis*) (2). This surface can be divided into medial and lateral facets, slightly separated by a reduced osseous crest. Overall, the surface is the negative of the articular facet of the incus to whom it articulates. 

In the ventral lateral direction, the head of the malleus continues with the second segment, the neck of the malleus (*Collum mallei*) (5), with an oblique placement. Slightly flattened in the cranial and caudal directions in the studied species, the neck of the malleus is quite a short segment between the head (1) and the handle of the malleus (7). In the initial segment in the ventral cranial part of the surface, we could identify in the studied individuals the presence of a relatively small anterior process (*Processus rostralis*) (3). In this portion, in the ventral and cranial part of the neck, close to the articular surface, the presence of a reduced bony tubercle was identified. The second process on the surface of the neck of the malleus, the lateral process (*Processus lateralis*) (4), was very well-developed in all individuals examined in this study. This process appears conical in shape, with a dorsolateral orientation, and at the level of the neck, it starts to slightly curve, continuing with the handle of the malleus (*Manubrium mallei*). The angulation between the handle and the neck in the malleus is acute.

The handle of the malleus in all the animals studied appeared as a long piece of bone with a triangular shape across its entire length except for in its distal part, where this flattened in the lateral and medial parts. This makes this terminal end resemble the shape of a spatula that is also slightly curved in the mediolateral plane. The lateral surface of the handle of the malleus makes contact with the tympanic membrane along its entire length. In this species, the handle of the malleus follows a parallel orientation to the long crus of the incus.

On the medial surface of the manubrium of malleus, the muscular process (*Processus muscularis*) (6) was noted as the presence of a reduced bony tubercle that serves as the insertion point for the tensor muscle of the tympanic membrane (*M. tensor tympani*).

Histologically, the non-articular part of the head of the malleus mainly comprises compact bony tissue, and the presence of some blood vessels can be noticed. The articular surface, covered with hyaline articular cartilage, is adjacent to a layer of compact subchondral bone that is highly vascularized. The articular capsule of the malleo-incal joint appears to be made from dense connective tissue. The histological investigation of the handle of the malleus revealed the existence of a medullary cavity filled with adipocytes. The wall of the handle shows the existence of compact subchondral bone covered with hyaline cartilage (Figure 2).

### 3.2. Incus

The *Incus* (Figure 3) is the second auditory ossicle and articulates with the malleus via its articular surface. Slightly reduced in size (Table 4), it matches the typical description of the ossicle as a biradicular molar with the following morphological parts: the body of the incus (*Corpus incudis*) (1), the long crus (*Crus longum*) (4), and the short crus (*Crus breve*) (3).

The body has an approximately rectangular shape with a visible malleal articular surface at its lateral extremity. This surface comprises a medial and a lateral facet, separated by a small groove that makes it resemble the concavity of a saddle. Opposite this articular surface, the ossicle continues along the two processes (crura) (3,4) that form an almost 90-degree angle. The difference between the sizes of the two crura (the long and the short) is noticeable, both morphologically and metrically.

The short crus (*Crus breve*) (3) that protrudes from the dorsal part of the ossicle is relatively short and almost conical in shape. This process is suspended from the ceiling of the tympanic cavity via the posterior incudal ligament (*Ligg incundis*).

The second process (the long crus) (*Crus longum*) (4) continues from the body in a parallel direction to the handle of the malleus (*Manubrium mallei*). This process is far longer than the short one and much thinner. Its distal terminal end appears elongated and slightly curved, continuing with the lenticular process (*Processus lenticularis*) (5), which is an oval, slightly flattened bony piece that serves to articulate with the stapes.

The histological investigation showed that the articular surface is covered with hyaline cartilages that cover a vascularized, compact subchondral bony layer (Figure 4). The articular capsule comprises dense connective tissue.

The short crus (*Crus breve*) (3) is made of compact bone with numerous blood vessels. At the distal extremity of the short crus, a mineralized cartilaginous area serves for the attachment of the caudal incal ligament (*Ligg incudis*) (comprising dense connective tissue). The long crus (*Crus longum*) (4) seems to be similarly structured with dense compact bone and is highly vascularized. Notable is the fact that within the compact layers of the long crus, a medullary cavity was identifiable that was filled with adipocytes, as revealed via our investigation. The articular surface of the lenticular process of the incus where it is nearest to the head of the stapes (the incudostapedial joint) (*articulatio incundospapedia*) is covered with calcified cartilage supported by compact subchondral bone (Figure 4).

### 3.3. Stapes

The stapes (*stapes*) (Figure 5) is the smallest of the ossicles (Table 5). It is the deepest-placed component of the assembly and articulates with the incus via the lenticular process adjacent to the oval window (*Fenestra vestibuli*).

The stapes has an overall trapezoidal shape made from the head (*Caput stapedis*) (2) and two crura, an anterior (*Crus rostrale*) (3) and a posterior (*Crus caudale*) (6) one, which are connected to the base (*Basis stapedis*) (7), which rests on the oval window (*Fenestra vestibuli*).

At the head of the ossicle, an ovoidal articular surface with a slight central excavation is visible. This articular surface connects the stapes with the lenticular process of the incus at the level of the long crus of the ossicle.

The caudal border of the head of the stapes appearing as a reduced tuberosity is noted that serves as an insertion point for the distal stapedial muscle (*M. stapedius*).

In the medial and caudal directions, the head of the stapes continues with the two crura that connect to the basis of the stapes, the footplate (7). The two crura have slightly divergent directions and delimit a circular space in the intercrusal foramen (8). Specifically, the posterior crus (6) is more straight and slightly thicker than the anterior one (3). In one specimen, a fragment of a bony spicule (Figure 5) was visible, which was determined to be a remnant of the stapedial artery [29].

The stapedial footplate (*basis stapes*) (7) is ellipsoid-shaped in all the individuals in this study, and its circumference has a noticeably thickened halo. The external surface of the footplate is slightly excavated, while the internal one has a wavy appearance, with a convex caudal part and a slightly convex part in its anterior third. The footplate allows the junction with the oval window utilizing the annular ligament (*Lig. Anulare stapedis*).

The histological investigation showed that the articular surface of the stapes is covered with hyaline cartilage with scant mineralized areas, and the articular capsule is represented as dense connective tissue. The tendon of the stapedial muscle was visualized as dense connective tissue on the caudal border of the head of the stapes (Figure 6).

The head of the stapes and the two crura are mainly composed of compact bony tissue, whereas hyaline cartilage and small, bilateral subchondral bone tissue were visualized within the footplate. The annular stapedial ligament appeared as a dense connective tissue between the footplate and the oval window (Figure 6).

## 4. Discussion

The present study aimed to reveal a series of morphological and morphometrical observations and descriptions of the auditory ossicles in the green African monkey.

The ossicular assembly within the tympanic cavity plays an important role in the auditory system. Despite its secure location within the temporal bone, any traumatic or pathological processes developing in the middle ear may debilitate or compromise the transmission of sound waves and impair normal hearing [25,30,31,32,33].

### 4.1. Gross Morphological Features

Similar to the reported data in the case of humans [5,21,26,28,34,35,36] or other primates [5,13,16,21,22,25,37,38,39], the ossicular assembly in *Chlorocebus aethiops sabaeus* consists of three well-individualized bony pieces located in the cavity of the middle ear (*auris media*), the most laterally placed and the largest being the malleus.

The morphological features of the malleus are similar to those described in the literature for *G.gorilla* și *G. beringei* [21], with some different features mentioned in *Pongo* sp., *P. troglodytes*, *P. paniscus*, and *S. syndactylus* și *H. sapiens* [26].

Another feature resembling one of the rhesus monkey [13] points to the neck of the malleus, which is very short in comparison with the very well-developed head. This contrasts with the observations described in most of the platyrrhine, wherein the neck of the malleus is completely absent as a describable segment [25].

There also seems to be a significant variation in the morphology of the handle of the malleus between related species. While this study observes the existence of an easily visible lateral process (*processus lateralis*) in the handle of the malleus with a pointed, conical aspect, similar to that in *Macaca mulatta* [13], a different degree of development is mentioned in prosymians and canarrhines which have been reported to have well-represented morphology of this process. The platyrrhines, in contrast, lack this process altogether [25,39].

The anterior process (*processus anterioris*), situated on the ventral and cranial malleal articular surface, was easily identified and well-developed in this study. It has been documented that this anatomical structure has the most diverse representation in the auditory ossicle prominences between species [25]. The presence of the anterior process seems to be specific to Old World primates [16], as it has this bony prominence during the developmental ontogeny of the individual [22,40]. For example, in humans, its size is reported to vary with age, and the observed size has been documented to regress in size as the individual gets older.

Specifically, it seems the most proportionately prominent phase of its development is described in fetuses, but it has also been observed in children as a very evident bony process. In contrast, it seems to regress and become more blunt and reduced in size in the adult tubercle [35]. A similar phenomenon has been noted for the visible anterior process in *Homo sapiens*, *Pan troglodytes*, *Pan paniscus*, *G. beringei*, *G. gorilla*, *Pongo* sp., and *S. syndactylus* by other authors [22,40].

This process acts as a rostral anchoring point of the ossicular assembly, facilitating rotational movement in the middle ear. The anterior process also serves as an insertion point for the anterior malleal ligament, a fact that is also reported in the *Macaca Mullata* species [13]. This series of observations indicates the likelihood of a free mobility type of ossicular assembly [41,42].

In some earlier studies, it has been stated that in the case of species with a reduced anterior malleal process, the handle of the malleus and the long crus of the incus are parallel and perpendicularly placed on the horizontal plane. This fact has been confirmed in the case of *Chlorocebus aethiops sabaeus* studied individuals [40].

The third malleal process (the muscular process, or *processus muscularis*) is on the medial surface of the manubrium, and in its median segment, serves as an insertion point for the tensor timpani muscle. Similar features have been reported in most other primates [13,17,19,25] and *Herpetes Javanicus* [43]. In contrast, the tensor tympani muscle seems to be absent in species from *Notoryctidae, Chrysochloridae, Tupaiidae,* and *Ochotonidae,* as well as most from the *Talpidae*, *Manidae,* and *Spalacidae* families [44].

Unlike the situation encountered in the green monkey, the literature describing *Callithrix jacchus* [45] as well as humans [19] notes the total absence of all three processes on the surface of the malleus.

Regarding the handle of the malleus, this section was observed as triangularly shaped with a wider lateral surface than its medial one. In comparison with *Macaca Mulatta,* the handle is more rounded in its cross-sectional description [13].

Our observation of a spatulated distal segment of the handle is similar to the data reported on *Macaca Mulatta* [13,19] and on other primates [21,22]; however, in *H. sapines*, the handle is shorter, and a large variability in the malleus is reported [34].

In the incus, like in humans and rhesus monkeys [13], this study confirms the presence of a biradicular molar with a long crus and a short one that is detached. The same stands in the case of the description of the incus in *Pan troglodytes*, *Gorilla gorilla* [5], and in humans, with the second ossicle being the most uniform from a morphological perspective [33,34,35]. The literature has placed special attention on the incus for its importance in normal functioning and its role in pathologic processes. Specifically, the long crus of this ossicle may be affected in cases of chronic otitis in the middle ears [36,46,47,48]. Another important fact associated with the morphology of this ossicle refers to the existence, in some particular instances, of the malleo-incal complex, which is the result of the fusion of the first two ossicles, noted in *Heterocephalus glaber* [49,50], *Cuniculus paca* [51], *Octodon degus* [52], guinea pigs [53,54], and chinchillas [32,54,55,56]. The malleo-incal complex is also implicated in specific cases of humans and mice as a probable explanation for deafness [53]. The total absence of the incus has also been cited in birds [33,57,58,59,60,61] and some reptiles [33,57,62].

This study described the long crus of the incus that terminates with an oval-shaped structure (the lenticular process, *processus lenticularis*), similar to that which is seen in other primates [13,17,21,22,25]. The shape of this process differs, as in *Macaca Mulatta*; however, in [13], this process was found to be shaped as a round structure rather than an oval one. In some other instances, this small terminal piece may be completely absent [32,63].

Given its anatomic and topographic placement and its describable features, the stapes in the green monkey seems to correspond to that which has been previously described in other primates. Our study confirms the results from other studies [5], which report the existence of a head and two crura that delimit the intercrusal foramen and the base.

As identified previously, the stapes is the least variable ossicle in terms of intra- or interspecific morphology [16], a fact that is reflected in the described situation for the *Chlorocebus aethiops sabaeus* specimens. One notable difference is seen in humans, wherein the highest degree of variability is reported, with several variances in morphological elements being detailed [35], each feature having surgical importance [32,64]. Moreover, a high degree of fragility is noted in this ossicle, as the stapes seems to undergo a progressive loss in density and continuous narrowing in width over time [65], a fact that makes adults more prone to mechanical injuries compared with younger individuals [25,65,66,67].

Overall, the appearance of the stapes (a frame with a trapezoidal geometrical shape) is similar to that in the macaque [13,19] and many other primates [17,21,22,25,45] including humans [21,22,34,35,65]. This shape also holds true for the guinea pig [68], hamster [69], and chinchilla [31]. This trapezoidal shape seems to be relatively ubiquitous; however, in the human fetus, the overall shape of the stapes varies and is more rectangular in its presentation [70].

The reduced muscular process on the caudal part of the stapes is similar to that in the macaque [13]. In both species, this process serves as an insertion site for the stapedial muscle, unless it belongs to some specific mammals wherein this muscle does not exist. [44]. In humans, the literature describes the flattening of this process to a simply rough surface [64,71,72]. In some adult primates, a bony spicule has been identified in which the stapedial artery regresses [17,25,73]. A similar manifestation of this entity is detailed in studies that focused on the human fetus [32,49,50,74]. The footplate of the stapes in our specimens took on an almost ellipsoidal shape, which is slightly different from the situation described in humans [22], wherein a kidney-shaped footplate is noted.

### 4.2. Micromorphological Features

Available sources are scarce regarding data related to the histology of the auditory ossicles in monkeys. Some studied materials cite a large variability in bone density [25], with little or no specific reference.

Specifically, in the malleus, the data reported are almost entirely focused on human specimens, pointing to a structure of compact bone tissue covered in some parts with hyaline connective tissue and the presence of arterial capillaries as well as a defined medullary cavity [75]. The findings of this study show largely mirrored descriptions between our specimens and humans.

The incus is also composed of compact bony tissue, covered at the articular surface with hyaline cartilaginous tissue, and it is quite similar to the description noted in humans [36,75,76]. This study detailed the incus containing significant capillary vasculature and cavitary spaces filled with adipose tissue embedded in the bony tissue.

The malleo-incal joint, which may be among the oldest joints from a phylogenetical perspective [65], was seen as an articular capsule comprising dense connective tissue in our specimens.

The histological features of the stapes confirm that among the three ossicles, this one is the most delicate. The structural features identified herein are very similar to the ones found in the previous literature describing human ossicles [25,34].

### 4.3. Morphometrical Interpretation of Data

Our efforts attempted to document the variability that was noted between individuals, specifically with correlation to sex and the side of the body from which the tissue originated.

Before the in-depth statistical approach, a series of values were used to check the normality of the data. Most of the data (*n* > 100) originated from the measurements of the malleus, while the Kolmogorov–Smirnov test of normality was the basic choice to check the series of measurements of the total length of the malleus (Table 6) (measurement no. 1; *n* = 146; *p*-value of 0.08314; σ = 0.1724; df = 146), manubrium thickness (measurement no. 2; *n* = 146; *p*-value of 0.24378; σ = 0.05629; df = 145), neck width (measurement no. 6; *n* = 146; *p*-value of 0.48663; σ = 0.05349; df = 145), and head width (measurement no. 7; *n* = 146; *p*-value of 0.22077; σ = 0.1248; df = 145). For the angle values, where *n* = 47, the normality test indicated again the validity of the series as a normal distribution (*p*-value of 0.22409) (see Table 1 for the measurements). For the metrical data on the incus, we checked the long process length (measurement no. 10; *n* = 38; *p*-value of 0.28363), the short process length (measurement no. 9; *n* = 38; *p*-value of 0.99759), the functional length of the long process values (measurement no. 11; *n* = 38; *p*-value of 0.92986), and the angle between axes values (measurement no. 14; *n* = 38; *p*-value of 0.73952) (as designated in Table 2). The data from the stapes measurements we used were the total height of the stapes (measurement no. 19; *n* = 17; *p*-value of 0.27329), and the footplate length (measurement no. 31; *n* = 31; *p*-value of 0.20137) (see Table 3). For the nominated measurements, there is no indication of non-normal distribution, based on all the calculated *p*-values.

As a statistical procedure was used, a one-tailed paired or two-tailed unpaired t-test was performed to determine significant differences between values. For all statistical tests, we set the significance level to a *p*-value of less than 0.05. To reduce calculation errors and misinterpretation, we utilized only the data sets wherein the number of measurements was higher than 100 items.

A significant difference was noted between the values of the total length of malleus in males and females (Table 7). Specifically, the statistically significant difference between the sexes represented less than 3% of the values in males and females. No significant difference was found between the lengths of the ossicles when we attempted to separate or differentiate the left from the right side of the body. An interesting situation was observed regarding the thickness of the manubrium, wherein an 8% increase in size was noted in the mean values in males compared with females. A similarly significant difference was noted between the sexes regarding the head of the malleus width, with the mean value for males being about 4% higher than that for females. In these regards, our findings are consistent with other published data related to the sexual dimorphism seen in humans and African apes [5], which describe low levels of sexual dimorphism in ear ossicle size compared with body mass dimorphism between the sexes. A similar statement seems to be valid for data relating to body-side differentiation.

The number of available measurements for the incus was too small (within the range of 20–30 measurements each) to provide a reliable demonstration of sex differentiation. The values of the indices for the incus, as suggested in the literature data [5], are listed in Table 8 (Table 8).

### 4.4. Middle Ear Lever Ratio

Generally, the lever ratio compares the functional lengths of the malleus and incus based on their rotational axis inside the tympanic cavity. This assembly plays a key role not only in the transmission of vibrations but also in providing mechanical advantage, thus being a relevant physiological variable in the modeling of audition [5,77,78]. In this way, the authors of the aforementioned papers propose a standard set of measurements that characterize in detail the necessary measurements of the malleus and incus, based on the estimation of the rotational axis of the malleo-incal joint.

The functional length of the malleus (considered to be the manubrium length) and the length of the incus (considering the functional length of the long process) are the reference values for the calculation of the lever ratio as a fraction between these two (Table 9) [5]. As visible in Table 10 and Table 11, the computed values of the lever ratio for each individual (based on the average values for the individuals in which all necessary metric data were available) show no noticeable variation for the individuals, and the overall average values indicate a narrow 1.23 to 1.42 interval.

These values seem to be similar to values previously measured in humans, chimpanzees (*Pan troglodites*), gorillas (*Gorilla gorilla*) [5], and the machaca (*Machacus rhesus*) [18], which are all situated above the 1.4 value.

A possible explanation is described in a theory suggested by R. Quam, who hypothesizes that a lower value of this lever ratio is achieved through a shortening of the malleal manubrial part and the elongation of the functional length of the incus, which developed through evolution at the same time as a decrease in body size was occurring. This would explain the higher values for gorillas but does not explain the values for smaller species (such as macaques). The elongation of the incus (and its functional importance) is likely linked to a common ancestor in the evolutionary process [5], but this does not fully explain the situation mentioned earlier. Another attempt to correlate these lever-ratio data might also be inferred when judging the hearing ranges of the target species [79,80,81,82]. Although several factors are involved in the modulation of sound waves along the ossicular chain, the audible spectrum data currently available in different sources do not completely reflect the spectrum of values of the lever ratio.

## 5. Conclusions

The present study outlined and documented a series of morphological features of the auditory ossicles in the green monkey never before described in the scientific literature to the best of our knowledge.

The malleus of the green monkey consists of compact bony tissue covered with hyaline cartilage on the articular surfaces with a short but visible neck. The pointed lateral process is noted along with its long anterior process. The existence of a muscular process is also well demonstrated. Dimensionally, this bone shows little to no sexual and side dimorphism and has a functional average length (3.049 mm), which was targeted for the comparison between the lever ratios among a series of species, pointing to a situation similar to that which is described in humans.

We found the incus to consist of richly irrigated compact bone tissue and present an evident lenticular process situated at the terminal part of the long crus. Its functional length values (with an average of 2.31 mm) were compared with the lever ratio of 1.6 (average) and then compared with known values to establish a comparative basis for further studies in the field of otology.

Our research documents the stapes framed in a trapezoidal-shaped structure comprising compact boney tissue with surrounding hyaline cartilage on the articular surfaces. It was also observed to have a reduced muscular process and an ellipsoidal-shaped footplate that has an average surface area of 0.959 mm^2^ (0.891–1.05 mm).

## Figures and Tables

**Figure 1 biology-12-00631-f001:**
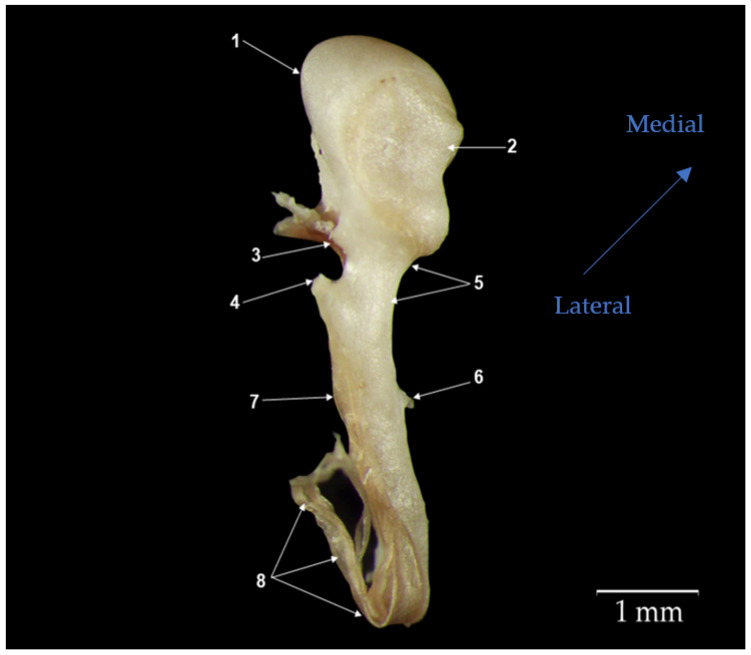
Anatomical features of malleus. (1) Head of malleus (*Caput mallei*); (2) articular surface (*Facies articularis)*; (3) anterior process (*Processus rostralis*); (4) lateral process (*Processus lateralis*); (5) neck of malleus (*Collum mallei*); (6) muscular process (*Processus muscularis*); (7) handle of malleus (*Manubrium mallei)*; (8) tympanic membrane fragments (*Membrana tympani*).

**Figure 2 biology-12-00631-f002:**
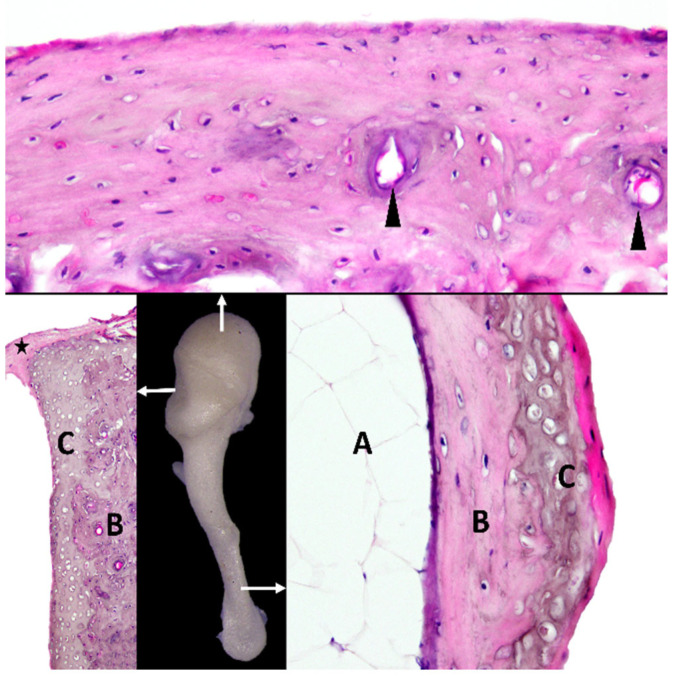
Histological aspects of malleus. Arrows indicate areas selected for histology. A—marrow cavity with adipocytes; B—subchondral compact bone; C—hyaline cartilage; arrowheads—blood vessels; star—incudomalleolar joint capsule with dense regular connective tissue.

**Figure 3 biology-12-00631-f003:**
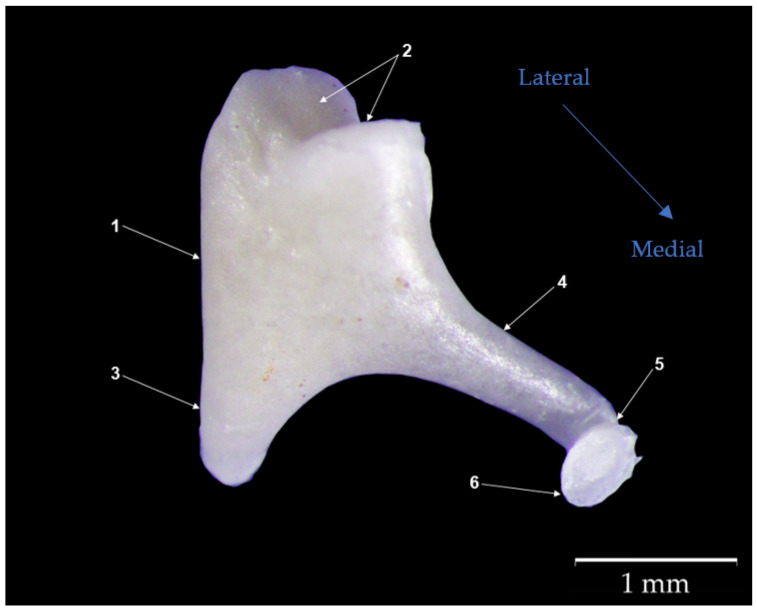
Anatomical features of the incus. (1) body of the incus (*Corpus incudis*); (2) malleal articular surface; (3) short crus (*Crus breve*); (4) long crus (*Crus longum*); (5) lenticular process (*Processus lenticularis*); (6) articular surface for the stapes.

**Figure 4 biology-12-00631-f004:**
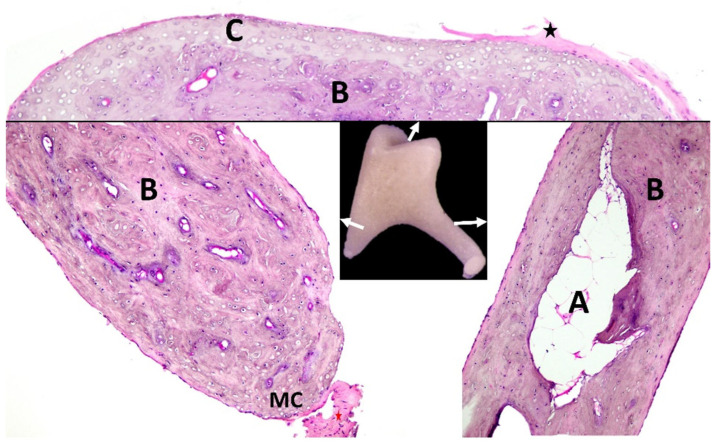
Histological aspects of the incus. Arrows indicate areas selected for histology. A—marrow cavity filled with adipocytes; B—compact bone; C—articular hyaline cartilage; black star—incudomalleolar ligament; MC—mineralized cartilage with posterior incudal ligament (red star).

**Figure 5 biology-12-00631-f005:**
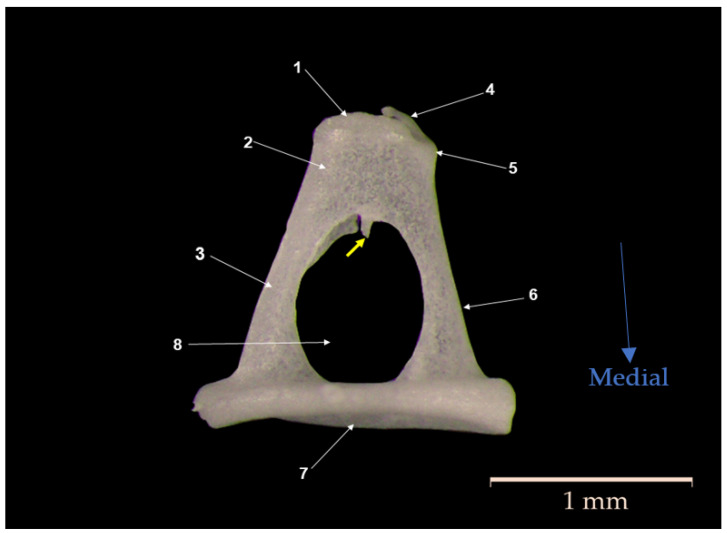
Anatomical features of the stapes. (1) articular surface for incus; (2) head of the stapes (*Caput stapedis*); (3) anterior crus (*Crus rostrale*); (4) tendon of the stapedial muscle; (5) stapedial muscle tubercle; (6) posterior crus (*Crus caudale*); (7) stapedial footplate (*Basis stapedis*); (8) intercrusal foramen; yellow arrow—the bony spicule, a remnant of the stapedial artery.

**Figure 6 biology-12-00631-f006:**
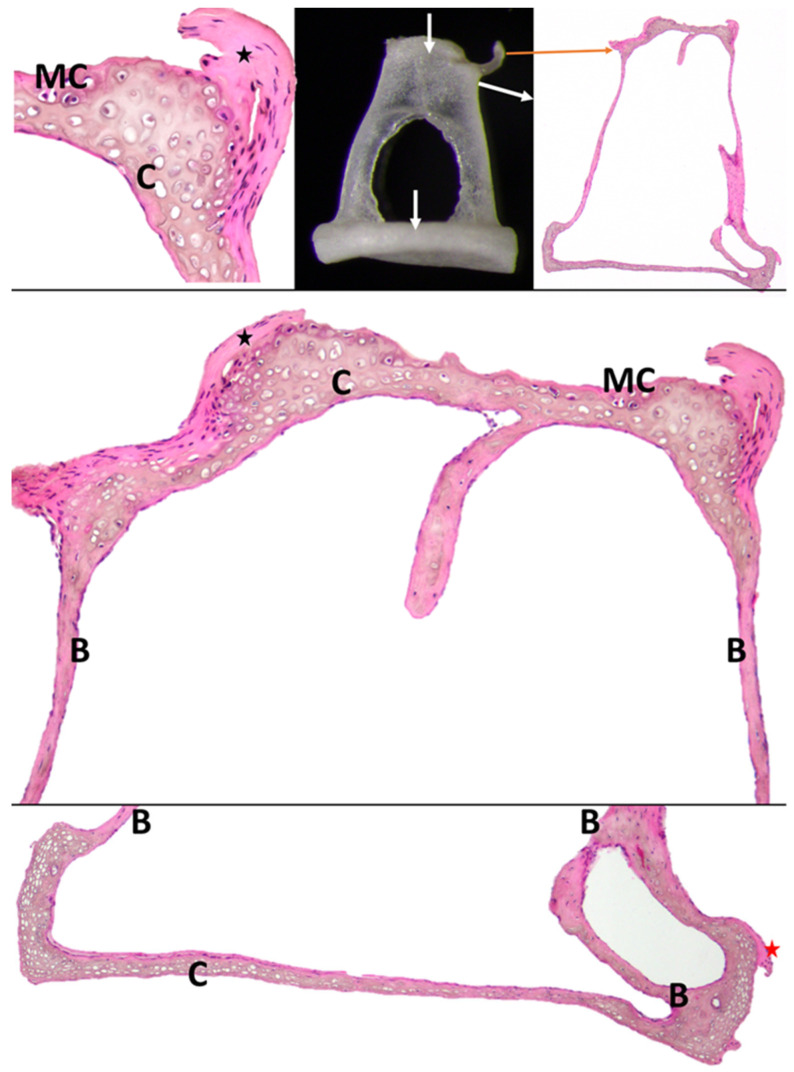
Histological aspects of the stapes. B—compact bone; C—articular hyaline cartilage; MC—mineralized cartilage; orange arrow—m. stapedius; black star—incudostapedial joint ligament; red star—annular ligament (*Ligamentum anularis stapedis*).

**Table 1 biology-12-00631-t001:** Measurement protocol and indices for the malleus [5].

*X*-axis	Midpoint of the minimum neck width—the most noticeable point along the top of the head
*Y*-axis	Most inferior point of the short process and the manubrium
1. Total length	Tip of the manubrium to the top of the head
2. Manubrium length	Tip of the short process to the tip of the manubrium following *X*-axis
3. Manubrium M-L thickness	M-L thickness of the manubrium at mid manubrium length, perpendicular to *X*-axis
4. Manubrium arc depth	Maximum depth of the curvature of the arc of the manubrium, following *X*-axis
5. Corpus length	Tip of the head to the lower border of the manubrium following *X*-axis
6. Neck width	Anterior and posterior borders of the neck
7. S-L head width	Maximum distance between 2 parallel lines marking the widest points of the margin of the head, taken following the *X*-axis
8. Angle between axes	X-Y angle
Manubrium/length index	(Manubrium length/total length) × 100
Manubrium robusticity index	(Manubrium ML thickness/corpus length) × 100
Manubrium/corpus index	(Manubrium length/corpus length) × 100
Corpus/length index	(Corpus length/total length) × 100

**Table 2 biology-12-00631-t002:** Measurement protocol and indices for the incus [5].

*X*-axis	Line that joins the most salient point along the anterior portion of the superior border of the body
*Y*-axis	Line that joins the tip of the long process to the most salient point along the superior border of the body
*Z*-axis	Line joining the tip of the long process to the most external point along the margin of the anterior facet
9. Short process length	Maximum distance from the tip of the short process to the most salient point along the anterior portion of the superior border of the body, following *X*-axis
10. Long process length	Maximum distance from the tip of the long process to the most salient point along the superior border of the body
11. Functional length of the long process	Perpendicular distance from the *Z*-axis (rotational axis) to the tip of the long process
12. Arc depth of the long process	Maximum depth of the arc along the long process measured from the plane defined by the lateral outmost point along the tip of the long process
13. Articular facet height	Max height of the articular facet with the bone oriented along the rotational axis
14. Angle between the axes	Angle formed by the *X*- and *Y*-axes
15. Interprocess length	Maximum distance between the most salient points along the superior margin of the short process and the tip of the long process
16. Interprocess arc depth	Maximum depth of the curvature between the short and long process tips
Incudal index	9/10 × 100
Long process index	11/10 × 100
Relative articular facet height	13/10 × 100

**Table 3 biology-12-00631-t003:** Measurement protocol and indices for the stapes [5].

*X*-axis	Line joining the antero-superior corner of the footplate and the tip of the head
*Y*-axis	Line joining the posterior superior corner of the footplate and the tip of the head
*Z*-axis	Line joining the most inferior points along the footplate margin anteriorly and posteriorly
19. Total height of the stapes	Maximum height from the lower margin of the footplate to the tip of the head perpendicular to the *Z*-axis
20. Head height	Minimum distance between the superior margin of the obturator foramen and the top of the head, measured perpendicular to the *Z*-axis
21. Obturator foramen height	Maximum height of the obturator foramen measured perpendicular to the *Z*-axis
22. Obturator foramen width	Maximum width of the obturator foramen measured parallel to the *Z*-axis
23. Maximum width of the crura	Maximum width across the anterior and posterior crura, measured on the external aspect and parallel to the *Z*-axis
24. Posterior crus length	Maximum distance from the posterior superior corner of the footplate to the tip of the head, following *Y*-axis
25. Posterior crus arc depth	Maximum depth of the curvature of the posterior crus measured parallel to the *Y*-axis
26. Anterior crus length	Maximum distance from the antero-superior corner of the footplate to the tip of the head following *X*-axis
27. Anterior crus arc depth	Maximum depth of the curvature of the anterior crus measured parallel to the *X*-axis
28. Angle A	Angle between the anterior and posterior crura or between the *X*- and *Y*-axes
29. Angle B	Angle between the anterior crus and the footplate or between the *X*- and *Z*-axes
30. Angle C	Angle between the posterior crus and the footplate between *Y*- and *Z*-axes
31. Footplate length	Maximum length of the footplate
32. Footplate width	Maximum width of the footplate
33. Footplate area	Measured area of the footplate
Stapedial index	31/19 × 100
Relative head height	20/19 × 100
Obturator foramen index	21/22 × 100
Footplate index	31/32 × 100
Crural index	36/24 × 100

**Table 4 biology-12-00631-t004:** Metric dimensions of the incudes in green monkey individuals.

Specimen	Measurement Number [5]	Mean Values—Right Side	Mean Values—Left Side	Mean Values
M1	9	2.475 mm		2.475 mm
	10	3.23 mm		3.23 mm
	11	2.18 mm		2.18 mm
	12	2.56 mm		2.56 mm
	13	1.025 mm		1.025 mm
	14	56.43°		56.43°
	15	2.445 mm		2.445 mm
	16	0.82 mm		0.82 mm
M2	9	2.68 mm	2.51 mm	2.57 mm
	10	3.37 mm	3.24 mm	3.28 mm
	11	2.21 mm	2.16 mm	2.18 mm
	12	2.56 mm	2.44 mm	2.48 mm
	13	1.18 mm	1.14 mm	1.15 mm
	14	49.8°	52.54°	51.6°
	15	2.55 mm	2.44 mm	2.47 mm
	16	0.8 mm	0.68 mm	0.75 mm
M3	9	2.47 mm	2.39 mm	2.40 mm
	10	2.95 mm	3.014 mm	3.0 mm
	11	2.13 mm	2.06 mm	2.07 mm
	12	2.38 mm	2.38 mm	2.38 mm
	13	1.12 mm	1.18 mm	1.17 mm
	14	46.1°	56.8°	55.08°
	15	2.09 mm	2.42 mm	2.36 mm
	16	0.78 mm	0.70 mm	0.71 mm
M4	9	2.36 mm	2.59 mm	2.42 mm
	10	3.18 mm	3.175 mm	3.18 mm
	11	2.21 mm	2.05 mm	2.17 mm
	12	2.57 mm	2.445 mm	2.54 mm
	13	1.31 mm	1.19 mm	1.28 mm
	14	52.53°	54.49°	53.02°
	15	2.38 mm	2.43 mm	2.39 mm
	16	0.69 mm	0.68 mm	0.69 mm
M5	9	2.72 mm	2.56 mm	2.61 mm
	10	3.45 mm	3.36 mm	3.39 mm
	11	2.23 mm	2.20 mm	2.21 mm
	12	2.43 mm	2.46 mm	2.45 mm
	13	1.35 mm	1.28 mm	1.30 mm
	14	55.13°	53.56°	54.08°
	15	2.78 mm	2.54 mm	2.62 mm
	16	0.59 mm	0.65 mm	0.63 mm
M6	9		2.61 mm	2.61 mm
	10		3.37 mm	3.37 mm
	11		2.31 mm	2.31 mm
	12		2.61 mm	2.61 mm
	13		1.54 mm	1.54 mm
	14		58.45°	58.45°
	15		2.65 mm	2.65 mm
	16		0.74 mm	0.74 mm

**Table 5 biology-12-00631-t005:** Metric dimensions of the stapes in green monkey individuals.

Specimen	Measurement Number [5]	Mean Values—Right Side	Mean Values—Left Side	Mean Values
M2	19	1.54 mm		1.54 mm
	20	0.61 mm		0.61 mm
	21	0.93 mm		0.93 mm
	22	0.74 mm		0.74 mm
	23	1.31 mm		1.31 mm
	24	1.77 mm		1.77 mm
	25	0.22 mm		0.22 mm
	26	1.82 mm		1.82 mm
	27	0.22 mm		0.22 mm
	28	59.05		59.05
	29	61.22		61.22
	30	56.3		56.3
	31	1.81 mm		1.81 mm
	32	0.722 mm		0.722 mm
	33	0.904 mm^2^		0.904 mm^2^
M3	19	1.54 mm		1.54 mm
	20	0.61 mm		0.61 mm
	21	0.93 mm		0.93 mm
	22	0.74 mm		0.74 mm
	23	1.31 mm		1.31 mm
	24	1.77 mm		1.77 mm
	25	0.22 mm		0.22 mm
	26	1.82 mm		1.82 mm
	27	0.22 mm		0.22 mm
	28	59.05		59.05
	29	61.22		61.22
	30	56.3		56.3
	31	1.81 mm		1.81 mm
	32	0.722 mm		0.722 mm
	33	0.904 mm^2^		0.904 mm^2^
M4	19		1.65 mm	1.65 mm
	20		0.64 mm	0.64 mm
	21		1.01 mm	1.01 mm
	22		0.806 mm	0.806 mm
	23		1.35 mm	1.35 mm
	24		1.83 mm	1.83 mm
	25		0.18 mm	0.18 mm
	26		1.876 mm	1.876 mm
	27		0.176 mm	0.176 mm
	28		58.93	58.93
	29		62.26	62.26
	30		60.36	60.36
	31		1.853 mm	1.853 mm
	32		-	-
	33			
M5	19	1.65 mm		1.65 mm
	20	0.673 mm		0.673 mm
	21	0.976 mm		0.976 mm
	22	0.793 mm		0.793 mm
	23	1.253 mm		1.253 mm
	24	1.856 mm		1.856 mm
	25	0.213 mm		0.213 mm
	26	1.863 mm		1.863 mm
	27	0.193 mm		0.193 mm
	28	53.96		53.96
	29	63.83		63.83
	30	58.96		58.96
	31	1.653 mm		1.653 mm
	32	0.813 mm		0.813 mm
	33	0.974 mm^2^		0.974 mm^2^
M6	19	1.595 mm	1.68 mm	1.658 mm
	20	0.685 mm	0.765 mm	0.738 mm
	21	0.91 mm	0.925 mm	0.92 mm
	22	0.79 mm	0.79 mm	0.79 mm
	23	1.36 mm	1.29 mm	1.315 mm
	24	1.845 mm	1.92 mm	1.896 mm
	25	0.26 mm	0.227 mm	0.238 mm
	26	1.975 mm	2.045 mm	2.02 mm
	27	0.235 mm	0.245 mm	0.241 mm
	28	55.3	51.46	52.74
	29	61.9	62.1	62.11
	30	62.69	64.6	63.96
	31	1.669 mm	1.684 mm	1.678 mm
	32	0.83 mm	0.813 mm	0.82 mm
	33	1.03 mm^2^	0.975 mm^2^	0.997 mm^2^

**Table 6 biology-12-00631-t006:** Metric dimensions of the mallei in green monkey individuals.

Specimen	Measurement Number	Mean Values—Right Side	Mean Values—Left Side	Mean Values
M1	1	4.73 mm	4.86 mm	4.79 mm
	2			
	3	0.39 mm	0.43 mm	0.41 mm
	4			
	5			
	6	0.55 mm	0.54 mm	0.54 mm
	7	1.63 mm	1.60 mm	1.60 mm
	8	-	-	-
M2	1	4.84 mm	4.76 mm	4.81 mm
	2	3.02 mm	2.96 mm	3.01 mm
	3	0.35 mm	0.39 mm	0.36 mm
	4	0.26 mm	0.26 mm	0.26 mm
	5	3.38 mm	3.10 mm	3.35 mm
	6	0.58 mm	0.55 mm	0.57 mm
	7	1.26 mm	1.30 mm	1.27 mm
	8	149.7°	149.7°	149.7°
M3	1	4.66 mm	4.64 mm	4.65 mm
	2			
	3	0.40 mm	0.43 mm	0.41 mm
	4			
	5			
	6	0.57 mm	0.56 mm	0.56 mm
	7	1.34 mm	1.40 mm	1.37 mm
	8			
M4	1	4.94 mm	4.94 mm	4.94 mm
	2	2.86 mm	2.89 mm	2.88 mm
	3	0.40 mm	0.40 mm	0.40 mm
	4	0.27 mm	0.29 mm	0.28 mm
	5	3.01 mm	3.09 mm	3.05 mm
	6	0.58 mm	0.56 mm	0.57 mm
	7	1.43 mm	1.36 mm	1.39 mm
	8	145.1°	150.7°	148.12°
M5	1	4.84 mm	4.93 mm	4.88 mm
	2	3.14 mm	3.14 mm	3.14 mm
	3	0.38 mm	0.45 mm	0.41 mm
	4	0.30 mm	0.26 mm	0.28 mm
	5	3.05 mm	2.91 mm	2.99 mm
	6	0.55 mm	0.55 mm	0.55 mm
	7	1.33 mm	1.35 mm	1.34 mm
	8	152.73°	145.3	149.56°
M6	1	4.99 mm	5.07 mm	5.03 mm
	2		3.15 mm	3.15 mm
	3	0.40 mm	0.39 mm	0.39 mm
	4		0.18 mm	0.18 mm
	5		3.79 mm	3.79 mm
	6	0.55 mm	0.54 mm	0.55 mm
	7	1.43 mm	1.52 mm	1.48 mm
	8	-	154.13°	154.13°

**Table 7 biology-12-00631-t007:** Statistical results for the main measurements of the malleus (paired *t*-test for right and left and unpaired for male and female comparison).

Measurement No.	Right from Left	Male vs. Female
1	t = −1.032; *p* = 0.151901 The result is not significant at *p* < 0.05 Mean value for right = 4.85 (*n* = 81); mean value for left = 4.55 (*n* = 65)	t= −2.0770; *p* = 0.039581 The result is significant at *p* < 0.05 Mean value for females = 4.81; mean value for males = 4.88
3	t = −3.10126; *p* = 0.001161 The result is significant at *p* < 0.05 Mean value = 0.38 (*n* = 80); mean value for left = 0.41 (*n* = 65)	t= 2.077; *p* = 0.00196 The result is significant at *p* < 0.05 Mean value for males = 0.4 (*n* = 113); mean value for females = 0.37 (*n* = 32)
6	t = 1.45139; *p* = 0.074431 The result is not significant at *p* < 0.05 Mean value for right = 0.57 (*n* = 81); mean value for left = 0.55 (*n* = 64)	t = −0.57247; *p* = 0.283953 The result is not significant at *p* < 0.05 Mean value for males = 0.56 (*n* = 112); mean value for females = 0.57 (*n* = 33)
7	t = −1.279; *p* = 0.101485 The result is not significant at *p* < 0.05 Mean value for right = 1.37 (*n* = 80); mean value for left = 1.39 (*n* = 65)	t = 2.15017; *p* = 0.016612 The result is significant at *p* < 0.05 Mean value for males = 1.39 (*n* = 113); mean value for females = 1.34 (*n* = 32)

**Table 8 biology-12-00631-t008:** Variance indices for the incus [5].

Index	Value (Average/Variation)
Incudal index	77.76 (69.7–87.16)
Long process index	67.30 (61.3–86.9)
Relative articular facet height	38.66 (30.69–50.33)

**Table 9 biology-12-00631-t009:** Variance indices for the malleus [5].

Malleus Index	Value (Average/Variation)
Manubrium/length index	62.36 (57.9–72.4)
Manubrium robusticity index	12.19 (6.2–17.02)
Manubrium/corpus index	96.38 (77.88–118.32)
Corpus/length index	64.0 (53.5–87.7)

**Table 10 biology-12-00631-t010:** Variance indices for the stapes [5].

Index	Value (Average/Min and Max)
Stapedial index	108.67 (100–117)
Relative head height	41.48 (37.03–48.2)
Obturator foramen index	124.13 (106.4–138.23)
Footplate index	214.29 (176.7–270)
Crural index	102.16 (89.28–11.95)

**Table 11 biology-12-00631-t011:** The malleus/incus lever ratios for the green monkey individuals studied.

Specimen Number	Functional Length of Malleus	Functional Length of Long Process	Lever Ratio
M1	No available data	2.18 (*n* = 2)	
M2	3.01 (*n* = 12)	2.18 (*n* = 9)	1.38
M3	No available data	2.07 (*n* = 6)	
M4	2.88 (*n* = 16)	2.17 (*n* = 8)	1.32
M5	3.14 (*n* = 14)	2.21 (*n* = 9)	1.42
M6	3.15 (*n* = 3)	2.31 (*n* = 4)	1.36
Average value	3.045	2.31	1.3730

## Data Availability

The data that support the findings of this study are available from the corresponding author, G.A., upon reasonable request.

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
