# Peer review of "Anatomical, Histological, and Morphometrical Investigations of the Auditory Ossicles in Chlorocebus aethiops sabaeus from Saint Kitts Island"

_biology, 2023, doi:10.3390/biology12040631_

Round 1

Reviewer 1 Report

Dear authors,

I find your paper very interesting and valuable for researchers.

In addition to a few minor errors, the main complaint is the tables with the results of your own research in the Discussions chapter. Therefore, I am of the opinion that tables 4-11 should, with a few corresponding sentences of explanation, be placed in the appropriate subchapters in the Results (Tables 4-6 with the corresponding description should be inserted in the Results chapter under 3.1, tables 7 and 8 under 3.2, and tables 9-11 under 3.3).

In the Discussion chapter, only comparisons with other research and authors' observations should remain.

Author Response

Dear reviewer,

Thank you very much for your nice words and clear comments and suggestions that you made. Here you have a response to your notes:

Dear authors,

I find your paper very interesting and valuable for researchers.

  • In addition to a few minor errors, the main complaint is the tables with the results of your own research in the Discussions chapter. Therefore, I am of the opinion that tables 4-11 should, with a few corresponding sentences of explanation, be placed in the appropriate subchapters in the Results (Tables 4-6 with the corresponding description should be inserted in the Results chapter under 3.1, tables 7 and 8 under 3.2, and tables 9-11 under 3.3).

The tables were moved according to your suggestion

  • In the Discussion chapter, only comparisons with other research and authors' observations should remain.

By removing the tables from this part, the Discussion section contains only the data type mentioned by the reviewer

Reviewer 2 Report

The manuscript in question is well presented and motivated morphometrical and histological characterization of previously non-described auditory ossicles of the African green monkey. The manuscript goes into a lot of depth to fully describe and characterize the ossicles in detail and should be truly credited for doing so. However, the methods and results require more clarity, correction, and additional analyses prior to publication.

Please see directed feedback in attached document below:

The introduction, although brief due to the limited research available in the field, is well motivated and argues for the potential utility of the present study to allow new research in the field of comparative otology, based on recent and relevant research.

The methods are appropriate for the type of study and rely on previously established measurement criteria. The following minor revisions should be considered to clarify some details of the methods and allow replicability of the study:

1.       Please outline the sample in a table indicating how many ear ossicles were used from each animal by sex and by side, giving specific n values for each group. This is important as some of the n values do not align with the sample as presented here. More on this under the results section.

2.       Please describe what was done to the images in Adobe photoshop. While I suspect that the authors would have probably only cleared and made the background uniform and perhaps improved brightness and contrast, it should still be described. Particularly since the measurements are taken from these images, minor differences could arise from different processing approaches applied in Photoshop as edges are less or more visible at different contrast levels.

3.       Please indicate how long the ossicles were fixed in 10% NBF, as prolonged fixation times can affect some staining techniques and induced biomechanical property changes in bone.

4.       Please clarify what was the desired level of decalcification (complete?) and please clarify how this was checked? (X-ray, chemical end point testing?)

5.       Please clarify what microtome and blades were used to cut the ossicles.

6.       Kindly refer to tables 1-3 in text when discussing the measurements of the ossicles in the methods. Perhaps include the citation for the paper describing the measurements in the table headings as well.

7.       From the manuscript it is not obvious if repeatability of the measurements was tested. Please include the procedure and statistical analysis for intra- and inter-observer error testing for the measurement data.

8.       Please include the procedure outlining the statistical analysis for the study.

The results and discussion are mostly well presented and organized in relation to the morphological description and the histological characterization. However, the measurements discussed in methods should be presented firstly in the results and not in discussion and some of the information on the measurements is not clear as presented. Please revise the following:

1.       Please add anatomical directionality compass in the figures of the individual ossicles displayed to help the reader orientate themselves while following the description in the text of the results.

2.       Please include the appropriate references to figure 1 in the text of the manuscript to help guide the reader.

3.       Please include all the anatomical terminology used in the text in the figure captions as well for consistency and clarity (e.g., processus rostralis as well as anterior process for figure 1, landmark 3).

4.       Lines 154 and 156: Please clarify if the anterior process as processus rostralis (line 154) and later on the bony tubercle as processus rostralis again (line 156) are the same feature or if this was intended to refer to a different structure.

5.       Line 285: Please clarify what is meant by “the entity”.

6.       Move the measurement data from the discussion section to the results section, after the morphology and histology have been described.

7.       Please refer to tables and figures in text when relevant, throughout the manuscript. It helps the reader follow the text and understand what the authors are trying to speak to.

8.       Please move the statistical analysis of the measurements in the results section as well.

9.       While I admire the transparency of including all measurements, please compile these in a supplementary material document and explain the missing values in that document (is there a difference between blank spaces or – spaces? Did these happen because the ossicles may have been partially damaged during extraction at the landmarks the measurements were using?).

10.   In addition, briefly present the descriptive statistics for the measurements (mean, standard deviation etc.). Presenting descriptive statistics allows for the ease of data visualization. It allows for data to be presented in a meaningful and understandable way, which, in turn, allows for a simplified interpretation of the data set in question. These can then briefly be interpreted and discussed in results and discussion respectively.

11.   Please make sure to also include the results of the repeatability tests of the measurements requested above.

12.   The statistical tests conducted are not justified and in part incorrect. A paired t-test can be conducted for left and right sides because they are the same individual and can be assumed to be dependent; however, when comparing dimensions between sexes, an unpaired t-test needs to be looked at as they are between different individuals. In addition, no normality or homoskedasticity has been presented/checked prior to selecting a t-test, which risks violating its assumptions. Please revise the statistics to avoid violating assumptions and invalidating potential results.

13.   Tables 5, 8, and 10: what is meant by “(average/variation)” in the column heading next to “Value”? Those terms have different meanings statistically, please confirm what was meant and amend appropriately.

14.   Lines 384 – 385: This is an interesting statement, please elaborate on the specific similarities.

15.   Please include a brief discussion on the bone marrow presence after discussing the individual bones. Especially in the context of human vs green monkey. Since to my knowledge human auditory ossicle bone marrow is in different location in the incus (body vs crus longus) and disappears by early childhood (Yokoyama T, Iino Y, Kakizaki K, Murakami Y. Human temporal bone study on the postnatal ossification process of auditory ossicles. Laryngoscope. 1999 Jun;109(6):927-30. doi: 10.1097/00005537-199906000-00016. PMID: 10369284.). In relation to this, it would be very interesting to also know the maturity of the animals used in the present study. Were they fully matured adult green monkey or still at some point of juvenile development? And if so at which point? Did you identify a difference in the marrow cavity size between the animals? Does this relate to age of the animals?

16.   Line 392: please amend mention of “genders” to “sexes”. The study looked at biological sex, not the socio-cultural implications of the term gender. While not necessarily distinct in green monkeys please refer to the more appropriate terminology for consistency with human primate research as well.

17.   Table 6:

a.       Please explain why some measurements are excluded from analysis (2, 4, 5, and 8).

b.       Is the mean for females for measurement 1 =4.88 or  (negative) –4.88?

c.       Why is “n=” so varied across the table? Some go as high as 113 others as low as 32 and I am unable to understand where these samples are coming from? The way the sample for the study is presented, it looks like there are 6 animals, with 2 sets of auditory ossicles each, making a total of 12 bones that could be measured. How are statistical analyses run on an n=113? Are measurements grouped and then compared for each other? If so, how, and why?

18.   Lines 396 – 399: this should be moved to methods and aligned with the description requested above to explain the statistical analysis procedure used for the study.

19.   Line 398 - 399: please clarify what is intended with “we have utilized only the data sets wherein the number of measurements was higher than 100 items”. Based on the available sample of 12 bones, each of which would have been measured once for each measurement number, this seems impossible.

20.   Please rephrase statement “Left Vs. Right”.

21.   Lines 424 – 429: are “the authors” referred to here the authors of the current manuscript or the authors of references 5, 79, and/or 80? Please clarify in text.

22.   Table 11: How is this ratio calculated? Please clarify and elaborate in the methods.

23.   Line 436: is “significant” indicating statistical significance or is it used qualitatively? If not of statical significance, please reword to avoid confusion and expectation of statistical significance. If statistical significance, please include results demonstrating this.

24.   Line 448: what is “illustrated below”? Should this perhaps be above? Please confirm and amend if necessary.

25.   Lines 448 – 452:  This is a very interesting point being made here. It would really enhance the manuscripts contribution elaborating on the findings of papers 81-84 regarding hearing ranges, which had honestly become a bit of an expectation when mention was made in methods of a functional auditory ossicle lever ratio. Please elaborate on where the current study’s data fits in the spectrum of hearing ranges of other primates based on the available data.

The conclusion overall sums up the manuscript well and highlights the major points of a novel study with new valuable information in the field of biology. However, little mention is made of its application in otology and possible future studies or other considerations. An attempt at this should be made by the authors.

Additional general comments:

Please review text for minor typographical and grammatical errors in the following line locations:

1.       183: spacing

2.       188: spacing

3.       261: bracket

4.       271: replace “si” with “and” and correct H. sapoiens

5.       310: remove italics from “as well as”

6.       311: italicize genus names.

7.       319: fix spacing, capitalization, and H. sapines

8.       354: amend “guineea”

9.       355: remove double period

10.   356: amend “varies is”

11.   461: amend “targeted” to “target”

Author Response

Dear reviewer,

Thank you very much for your time and observations. We have carefully analyzed your notes and corrected the mentioned issues.

Please see our responses below

The manuscript in question is well presented and motivated morphometrical and histological characterization of previously non-described auditory ossicles of the African green monkey. The manuscript goes into a lot of depth to fully describe and characterize the ossicles in detail and should be truly credited for doing so.

  • However, the methods and results require more clarity, correction, and additional analyses prior to publication.Please see directed feedback in attached document below:
  • The introduction, although brief due to the limited research available in the field, is well motivated and argues for the potential utility of the present study to allow new research in the field of comparative otology, based on recent and relevant research.
  • The methods are appropriate for the type of study and rely on previously established measurement criteria. The following minor revisions should be considered to clarify some details of the methods and allow replicability of the study:
  1. Please outline the sample in a table indicating how many ear ossicles were used from each animal by sex and by side, giving specific n values for each group. This is important as some of the n values do not align with the sample as presented here. More on this under the results section.

The ossicles were harvested from 6 individuals- 2 males and 4 females- moved this text section in the first line of the 2.1 section

  1. Please describe what was done to the images in Adobe photoshop. While I suspect that the authors would have probably only cleared and made the background uniform and perhaps improved brightness and contrast, it should still be described. Particularly since the measurements are taken from these images, minor differences could arise from different processing approaches applied in Photoshop as edges are less or more visible at different contrast levels.

Inserted in section 2.2 a brief statement about contrast and background processing

  1. Please indicate how long the ossicles were fixed in 10% NBF, as prolonged fixation times can affect some staining techniques and induced biomechanical property changes in bone

. Inserted in the Histology technique section-2 days fixation process

  1. Please clarify what was the desired level of decalcification (complete?) and please clarify how this was checked? (X-ray, chemical end point testing?)

Inserted in the Histology technique section- 1 hour slow (incomplete) decal followed by the surface decal. No other evaluations

  1. Please clarify what microtome and blades were used to cut the ossicles.

Inserted in the Histology technique section

  1. Kindly refer to tables 1-3 in text when discussing the measurements of the ossicles in the methods. Perhaps include the citation for the paper describing the measurements in the table headings as well.

Citation inserted in the maintext and in table captions

  1. From the manuscript it is not obvious if repeatability of the measurements was tested. Please include the procedure and statistical analysis for intra- and inter-observer error testing for the measurement data.

A brief note on basic repetability conditions is inserted in the 2.4 section, pointing to the main conditions of such a checkup. No interrater reliability checkup- as there was only one observer to perform the repeated measurements- was mentioned also in the maintext

  1. Please include the procedure outlining the statistical analysis for the study.

A line is dedicated to the description of the basic statistical procedures in section 4.3 (see lines 406-409) after the main section on measurements that also mentioned the statistic’s sources (line 126-129 on section 2.4)

The results and discussion are mostly well presented and organized in relation to the morphological description and the histological characterization. However, the measurements discussed in methods should be presented firstly in the results and not in discussion and some of the information on the measurements is not clear as presented. Please revise the following:

All tables were moved in the results section, as suggested

  1. Please add anatomical directionality compass in the figures of the individual ossicles displayed to help the reader orientate themselves while following the description in the text of the results.

Arrows indicating the general anatomical placements were inserted in all 3 figures

  1. Please include the appropriate references to figure 1 in the text of the manuscript to help guide the reader.

Each individual description now has a numeric reference to the captions of the figure. NAV names inserted in the captions, too

  1. Please include all the anatomical terminology used in the text in the figure captions as well for consistency and clarity (e.g., processus rostralisas well as anterior process for figure 1, landmark 3).

Each individual description now has a numeric reference to the captions of the figure. NAV names inserted in the captions, too

  1. Lines 154 and 156: Please clarify if the anterior process as processus rostralis(line 154) and later on the bony tubercle as processus rostralis again (line 156) are the same feature or if this was intended to refer to a different structure.

Corrected…basically it refers to same structure

  1. Line 285: Please clarify what is meant by “the entity”.

For clarity, I have replaced the term entity with “anatomical structure”

  1. Move the measurement data from the discussion section to the results section, after the morphology and histology have been described.

All tables have been moved from Discussion section to Results section

  1. Please refer to tables and figures in text when relevant, throughout the manuscript. It helps the reader follow the text and understand what the authors are trying to speak to.

Rechecked…made some more additions of reference to tables and figures

  1. Please move the statistical analysis of the measurements in the results section as well.

Tables of measurements and statistical aspects were moved into the suggested section

  1. While I admire the transparency of including all measurements, please compile these in a supplementary material document and explain the missing values in that document (is there a difference between blank spaces or – spaces? Did these happen because the ossicles may have been partially damaged during extraction at the landmarks the measurements were using?).

Explanation is simple- as measurements were taken repeatedly on the digital images, we have used the series of images of the same ossicle that were taken from different angulations…so in some images, some sets of measurements were impossible or non-viable..so I preferred to skip these relative measurements for the sake of repeatability and acuity. Some others were unavailable due to fragmentation and damage during the collection.No difference between empty and – spaces…corrected in tables and replaced.

  1. In addition, briefly present the descriptive statistics for the measurements (mean, standard deviation etc.). Presenting descriptive statistics allows for the ease of data visualization. It allows for data to be presented in a meaningful and understandable way, which, in turn, allows for a simplified interpretation of the data set in question. These can then briefly be interpreted and discussed in results and discussion respectively.

The methodological statistics were, as suggested by the other reviewer as well, stressed and highlighted on the initial section dedicated to methodology (section 2.4) and detailed a little in the 4.3 section that deals with the variation  and statistical data extracted from the statistical tests performed

  1. Please make sure to also include the results of the repeatability tests of the measurements requested above.

A brief note on basic repeatability conditions is inserted in the 2.4 section, pointing to the main conditions of such a checkup. No interrater reliability checkup- as there was only one observer to perform the repeated measurements- was mentioned also in the maintext

  1. The statistical tests conducted are not justified and in part incorrect. A paired t-test can be conducted for left and right sides because they are the same individual and can be assumed to be dependent; however, when comparing dimensions between sexes, an unpaired t-test needs to be looked at as they are between different individuals. In addition, no normality or homoskedasticity has been presented/checked prior to selecting a t-test, which risks violating its assumptions. Please revise the statistics to avoid violating assumptions and invalidating potential results.

Two tailed unpaired t-test was redone in case of the three main measurements of male and female. The results were corrected, but basically the similar output was available, with slightly different values for the 1-st measurement We highlighted the results of the normality test on some of the series we used in some calculations.we kept only data from significant series- more than 30 measurements. A dedicated paragraph in 4.3 section

  1. Tables 5, 8, and 10: what is meant by “(average/variation)” in the column heading next to “Value”? Those terms have different meanings statistically, please confirm what was meant and amend appropriately.

We did not mean the mean statistical meaning but to show the value intervals among the series. So we corrected the header with the min&max value

  1. Lines 384 – 385: This is an interesting statement, please elaborate on the specific similarities.

The statement in regards to incus points to this structure/segment as being the frailest and finest when it comes to describing its construction. Might be regarded as a statements that might be in conjunction with some similar statements about the way bony tissue is structured at the level of the human incus and stapes- a reduced amount of tissue at the midlevel of the bone compared to the much denser one at the level of the two crura in incus - and the cavitation degree and vascular channels is mentioned by our sources, but not backed up by us as we lack clear comparative data.

  1. Please include a brief discussion on the bone marrow presence after discussing the individual bones. Especially in the context of human vs green monkey. Since to my knowledge human auditory ossicle bone marrow is in different location in the incus (body vs crus longus) and disappears by early childhood (Yokoyama T, Iino Y, Kakizaki K, Murakami Y. Human temporal bone study on the postnatal ossification process of auditory ossicles. Laryngoscope. 1999 Jun;109(6):927-30. doi: 10.1097/00005537-199906000-00016. PMID: 10369284.). In relation to this, it would be very interesting to also know the maturity of the animals used in the present study. Were they fully matured adult green monkey or still at some point of juvenile development? And if so at which point? Did you identify a difference in the marrow cavity size between the animals? Does this relate to age of the animals?

As mentioned in the initial materials and methods section, all animals were mentioned as being mature. We lack precise age data as the main experiment was executed by other colleagues and the cephalic extremities were presented as fragments for this second study. The lacunae observed were mentioned as being filled with adipos tissue, no clear traces of bone marrow. Similar to the previous observation, where we used the term “delicate”, we tried to point to the apparent enlargement/relative cavitation that we could asses just subjectively

  1. Line 392: please amend mention of “genders” to “sexes”. The study looked at biological sex, not the socio-cultural implications of the term gender. While not necessarily distinct in green monkeys please refer to the more appropriate terminology for consistency with human primate research as well.

Made the replacement

  1. Table 6:
  2. Please explain why some measurements are excluded from analysis (2, 4, 5, and 8).

We considered that there are not a sufficient number of measurements available for the statistical assessment. There is an explanation in section 4.3 in regards to the choice of measurement data.

  1. Is the mean for females for measurement 1 =4.88 or  (negative) –4.88?

corrected with =

  1. Why is “n=” so varied across the table? Some go as high as 113 others as low as 32 and I am unable to understand where these samples are coming from? The way the sample for the study is presented, it looks like there are 6 animals, with 2 sets of auditory ossicles each, making a total of 12 bones that could be measured. How are statistical analyses run on an n=113? Are measurements grouped and then compared for each other? If so, how, and why?

As explained in the methodology section and also at an earlier point (8) data was taken repeatedly on digital images and some of the angulations allowed the entire series of measurements, some others did not.

  1. Lines 396 – 399: this should be moved to methods and aligned with the description requested above to explain the statistical analysis procedure used for the study.

Moved to methodology section

  1. Line 398 - 399: please clarify what is intended with “we have utilized only the data sets wherein the number of measurements was higher than 100 items”. Based on the available sample of 12 bones, each of which would have been measured once for each measurement number, this seems impossible.

It is about number of available measurements as the measurements were seriate on digital images- same as in point 17c and point 8

  1. Please rephrase statement “Left Vs. Right”.

Rephrased

  1. Lines 424 – 429: are “the authors” referred to here the authors of the current manuscript or the authors of references 5, 79, and/or 80? Please clarify in text.

Corrected

  1. Table 11: How is this ratio calculated? Please clarify and elaborate in the methods.

Mentioned in the text. It is a simple division in between the 2 measurements, as mentioned by the source paper

  1. Line 436: is “significant” indicating statistical significance or is it used qualitatively? If not of statical significance, please reword to avoid confusion and expectation of statistical significance. If statistical significance, please include results demonstrating this.

Not statistically expressed. In terms of subjective difference, we have noticed a visible numeric difference for the few values available. Changed with the word “noticeable”.

  1. Line 448: what is “illustrated below”? Should this perhaps be above? Please confirm and amend if necessary.

Corrected

  1. Lines 448 – 452:  This is a very interesting point being made here. It would really enhance the manuscripts contribution elaborating on the findings of papers 81-84 regarding hearing ranges, which had honestly become a bit of an expectation when mention was made in methods of a functional auditory ossicle lever ratio. Please elaborate on where the current study’s data fits in the spectrum of hearing ranges of other primates based on the available data.

Unfortunately, from our perspective, there is not a good set of data that can help us in correlating the hearing range data (that are very diverse as far as spectrum mentioned for each species in indicated sources and also other sources) with the lever ratio data that we obtained from our sample. This might be, as you mentioned a very interesting point of start for a much more in-depth analysis, but we feel like this asks for a much larger spectrum of metric and scientific data that way overpasses the initial scope of this paper.

The conclusion overall sums up the manuscript well and highlights the major points of a novel study with new valuable information in the field of biology. However, little mention is made of its application in otology and possible future studies or other considerations. An attempt at this should be made by the authors.

We agree with this final statement., but as mentioned in the previous response, the goal of this attempt was mainly morphology- macro and micro, backed by some metrical data.

Additional general comments:

Please review text for minor typographical and grammatical errors in the following line locations:

  1. 183: spacing
  2. 188: spacing
  3. 261: bracket
  4. 271: replace “si” with “and” and correct H. sapoiens
  5. 310: remove italics from “as well as”
  6. 311: italicize genus names.
  7. 319: fix spacing, capitalization, and H. sapines
  8. 354: amend “guineea”
  9. 355: remove double period
  10. 356: amend “varies is”
  11. 461: amend “targeted” to “target”

All taken in consideration and corrected

Round 2

Reviewer 2 Report

Dear Authors,

Thank you for addressing all queries and clarifying my concerns. Please include the information provided in your responses to reviewers in the methods text for the following points:

Response to point 8 regarding results; please include an explanation of how the measurements were taken at a vary basic point, even if this is based on a preexisting method from reference 5, the explanation of multiple measurements being captured from the various images at different angles and the missing ones being the ones that were not complete/able to be captured, should be included to not force the reader to secure access to the cited publication 5 in order to be able to understand the current paper's results.

Response to point 10 under results, the brief note on repeatability; please include information on standard deviation and degrees of freedom since a Type A Uncertainty repeatability was used.

In addition, please correct Table 6 under measurement 1 male vs female the reported p-value appears incorrect, also if it is indeed 0.395 then it is not significant as the line below says, please amend accordingly.

All the best.

Author Response

Dear Authors,

Thank you for addressing all queries and clarifying my concerns. Please include the information provided in your responses to reviewers in the methods text for the following points:

  • Response to point 8 regarding results; please include an explanation of how the measurements were taken at a vary basic point, even if this is based on a preexisting method from reference 5, the explanation of multiple measurements being captured from the various images at different angles and the missing ones being the ones that were not complete/able to be captured, should be included to not force the reader to secure access to the cited publication 5 in order to be able to understand the current paper's results.
    • Thanks for your suggestion. I have inserted a line with this basic explanation in Methodology section
  • Response to point 10 under results, the brief note on repeatability; please include information on standard deviation and degrees of freedom since a Type A Uncertainty repeatability was used.
    • We have included the sd data and df figures for the most important measurements (1,3,6,7) in the 4.3 section
  • In addition, please correct Table 6 under measurement 1 male vs female the reported p-value appears incorrect, also if it is indeed 0.395 then it is not significant as the line below says, please amend accordingly.
    • Thanks for the observation, the value in fact was 0.039581 (typo!), thus significant, as stated in the following lines in table 5

All the best.

Thank you very much for your patience and kind observations!